# Associations of Mental Health Issues with Health Literacy and Vaccination Readiness against COVID-19 in Long-Term Care Facilities—A Cross-Sectional Analysis

Linda Sanftenberg [1,*], Maresa Gschwendner [1], Andreas Grass [1], Marietta Rottenkolber [1], Isabel Zöllinger [1], Maria Sebastiao [2], Thomas Kühlein [2], Dagmar Hindenburg [3], Ildikó Gágyor [3], Domenika Wildgruber [4], Anita Hausen [4], Christian Janke [5], Michael Hoelscher [5], Daniel Teupser [6], Tobias Dreischulte [1], Jochen Gensichen [1] and on behalf of the BACOM Study Group [†]

1   Institute of General Practice and Family Medicine, LMU University Hospital, LMU Munich, 80336 Munich, Germany; maresa.gschwendner@med.uni-muenchen.de (M.G.); andreas.grass@med.uni-muenchen.de (A.G.)

2   Institute of General Practice, Friedrich-Alexander-University of Erlangen-Nuremberg, 91054 Erlangen, Germany

3   Department of General Practice, University Hospital Würzburg, 97080 Würzburg, Germany; hindenburg_d@ukw.de (D.H.)

4   Katholische Stiftungshochschule München, University of Applied Sciences, Campus Munich, Faculty of Health and Nursing, 81677 Munich, Germany; domenika.wildgruber@ksh-m.de (D.W.); anita.hausen@ksh-m.de (A.H.)

5   Division of Infectious Diseases and Tropical Medicine, Medical Center of the University of LMU Munich, 80802 Munich, Germany; christian.janke@lrz.uni-muenchen.de (C.J.); michael.hoelscher@med.uni-muenchen.de (M.H.)

6   Institute of Laboratory Medicine, LMU University Hospital, LMU Munich, 81377 Munich, Germany

*   Correspondence: linda.sanftenberg@med.uni-muenchen.de; Tel.: +49-89-4400-53388

†   Membership of the BACOM Study Group is provided in the Acknowledgments.

**Abstract:** Vaccinations against COVID-19 are of the utmost importance in long-term care facilities. During the pandemic, mental health issues increased significantly. This cross-sectional analysis aimed to assess the associations of depression and anxiety with health literacy in people in need of care and the association of depression and burnout with vaccination readiness against COVID-19 in health care workers (HCWs). Within our cross-sectional study, people in need of care were assessed for symptoms of depression (PHQ-9), anxiety (GAD-7), and health literacy (HLS-EU-Q16). Among HCWs, we assessed symptoms of depression (PHQ-9) and burnout (MBI-HSS), as well as psychological antecedents of vaccination (5C) to measure vaccination readiness against COVID-19. A multivariate regression analysis was performed. Symptoms of a major depression were significantly associated with reduced health literacy ($p = 0.010$) in people in need of care. Among HCWs, symptoms of depression and burnout reduced vaccination readiness against COVID-19 significantly. In particular, collective responsibility was reduced in HCWs suffering from burnout symptoms ($p = 0.001$). People in need of care and their HCWs could benefit from intensified target group-specific vaccination counseling. Additionally, more attention should be paid to the protection of mental health in long-term care facilities.

**Keywords:** people in need of care; healthcare workers; health literacy; vaccination readiness; mental health; COVID-19 pandemic

## 1. Introduction

The COVID-19 pandemic has led to repeated waves of infections and continues to pose a major challenge to society and healthcare systems worldwide [1]. People in need of care who often suffer from multi-morbidity, frailty and immune suppression, were particularly affected. Therefore, European nursing homes were locked down and

visitors were banned in Spring 2020. Many residents suffer from low to severe levels of cognitive impairment, which means that they do not understand the meaning of social isolation [2]. Health literacy extends beyond behavior-oriented communication and the concept of health education. It encompasses various determinants of health, including environmental, organizational, social and political [3]. For chronically ill individuals and people in need of care in particular, communication with healthcare workers (HCWs), the provision of health-related information and trust are necessary to increase health literacy [4]. Adequate health literacy is a basic requirement for obtaining and translating health-related information and has been shown to have a direct effect on health behaviors, including vaccinations [3,4]. Due to social restrictions during the pandemic, HCWs were in many cases the only personal contacts for people in need of care, as well as their only source of information [5]. Furthermore, HCWs were the most trusted advisors and influencers in regard to vaccination decisions [6].

Studies before 2020 did show that health literacy was affected in community-dwelling elderly people above 65 years of age who were suffering from depression and anxiety. Social interaction seems to have a protective effect [7]. Depression and anxiety could cause difficulties in understanding information about health and making decisions about preventive treatments [8]. During the pandemic, symptoms of these mental health issues increased dramatically, whereas social interaction was reduced [2,9–11]. Therefore, it can be hypothesized that symptoms of depression and anxiety were related to health literacy in German people in need of care during the COVID-19 pandemic.

HCWs who fought COVID-19 in long-term care facilities experienced a dramatic increase in mental health issues as well. They were exposed to high levels of stress and were at high risk for developing severe depressive symptoms and burnout [12,13]. Major depressive symptoms are associated with an increased risk of believing common misinformation about COVID-19 vaccines, which might correlate with vaccination readiness [10,14,15]. The association between burnout symptoms and vaccination readiness has not been elaborated in detail. Beyond mental health aspects, different psychological antecedents of vaccination have been identified to determine vaccination readiness in general. These are namely, trust in the safety and effectiveness of available vaccines and healthcare systems (Confidence), the perception of emerging risks of (non-)vaccination (Complacency), overcoming individual organizational barriers in everyday life (Constraints), the extent of active information seeking (Calculation) and a sense of social responsibility toward vulnerable groups (Collective Responsibility) [16].

HCWs caring for COVID-19-infected nursing home residents faced constant exposure to this virus [6]. In this context, vaccination was one of the most important preventive measures, fundamental for protecting both [17,18]. Nevertheless, vaccination rates remained poor in German long-term care facilities among residents and HCWs despite clear recommendations and easy access [19,20]. Since vaccinations are not mandatory in Germany, adequate health literacy and vaccination readiness are important pre-requisites to achieve sufficient vaccination rates. It can be hypothesized, that symptoms of depression and burnout were related to vaccination readiness in German HCWs during the COVID-19 pandemic.

Therefore, the aim of this study was to assess the association between symptoms of depression/anxiety and health literacy in people in need of care and the association between depression/burnout and vaccination readiness against COVID-19 in HCWs.

## 2. Materials and Methods

The reporting of this study follows the Strengthening the Reporting of Observational Studies in Epidemiology (STROBE) statement [21] (see Supplemental Materials: STROBE Statement—Checklist).

### 2.1. Study Design and Setting

This is a cross-sectional interim analysis based on data of the ongoing cohort study 'Bavarian ambulatory COVID-19 Monitor (BaCoM)', a dynamic prospective multicenter register in the State of Bavaria (Southern Germany) (German Register of Clinical Studies DRKS 26039) [22]. Study participants were recruited in inpatient (long-term care facilities) or outpatient care settings (home care provided by informal caregivers and/or outpatient care services). In order to maximize the geographical spread of study participants, we implemented a Bavarian-wide recruitment campaign with broad publicity.

The presented baseline data were collected from March 2020 to February 2023. Follow-up assessments were part of the cohort study (BaCoM) and were conducted at six-month intervals after baseline data collection for a period of up to three years in order to be able to observe the development of different clinical and mental health outcomes.

### 2.2. Participants

A purposive sample of up to 1000 people in need of care were recruited at three study sites in Bavaria (Munich, Erlangen and Würzburg). In addition, about 200 HCWs were recruited. People in need of care were identified via their general practitioner (GP), the long-term care facility they live in, via outpatient care services or informal caregivers, or via self-referral. Irrespective of how prospective people in need of care were identified, they were either enrolled by their GP or a study physician.

The GP recruitment was carried out within 240 GCP-qualified practices of the Bavarian Research Practice Network (BayFoNet) [23]. Additionally, eligible GPs with a past or current focus on managing patients with COVID-19 were identified. The participating GPs received compensation for their work within the study (participant inclusion and information, baseline examination, conducted surveys). For the recruitment of study participants (people in need of care and HCWs) from inpatient and outpatient care facilities, we used a list of about 700 eligible facilities in Bavaria with documented COVID-19 outbreaks (reporting system of the Bavarian State Office for Health and Food Safety).

#### 2.2.1. Eligibility Criteria for Patients

People in need of care or support were eligible if they receive financial support through public care insurance according to an officially assessed care level ("Pflegegrad") or a score of ≥5 on the 7-point Clinical Frailty Scale (CFS) [24]. Exclusion criteria were an estimated life expectancy of <6 months, missing health insurance, or unclear legal residency status.

#### 2.2.2. Eligibility Criteria for Participating HCWs

HCWs were eligible for recruitment if they are at least 18 years old and if they were employed in an outpatient or inpatient long-term care facility.

### 2.3. Variables

#### 2.3.1. Parameters of Interest among People in Need of Care

Socio-demographic data considered age, gender, ethnicity, marital status, education, and care-specific factors. To assess people in need of care, the Barthel Index was applied [25].

Symptoms of depression were measured using the Patient Health Questionnaire-9 (PHQ-9 score). This is a validated self-administered questionnaire consisting of nine items, each scoring one of the Diagnostic and Statistical Manual of Mental Disorders 5th edition (DSM-5) criteria for major depression with a sum score ranging from 0 to 27. A sum score of at least 10 indicates major depression. The items assess symptoms within the last two weeks with a Likert scale from 0 ("not at all") to 4 ("almost every day"). Sensitivity is reported to be 0.80 (95% confidence interval (CI) [0.71, 0.87]) and specificity to be 0.92 (95% CI [0.88, 0.95]) with a cut-off of 10 or higher [26].

The Generalized Anxiety Disorder-7 (GAD-7) questionnaire was used to measure anxiety disorders and their severity. The score ranges from 0 to 21, where a score of

≥10 is considered a suspected diagnosis of an anxiety disorder. The reliability coefficient Cronbach's alpha for the overall GAD-7 scale is 0.89 in the general public [27].

The level of individual evidenced health literacy was measured using the comprehensive Health Literacy Questionnaire (HLS-EU-Q16 (35]), which assesses subjective ability to understand information concerning one's health. This results in a classification based on the cut-off values of 1 to 8 = inadequate, 9 to 12 = problematic, and 13 to 16 = sufficient. Cronbach's alpha for HLS-EU-Q16 internal consistency is 0.89 [28].

### 2.3.2. Parameters of Interest among HCWs

Socio-demographic data considered age, gender, ethnicity, marital status and education. HCWs were asked about their function in the respective long-term care facility, type of care, their type of employment, and if they care for COVID-19-infected patients.

Again, the PHQ-9 was evaluated to assess the severity of depressive symptoms [26].

Furthermore, burnout symptoms were assessed using the Maslach Burnout Inventory Human Services Survey (MBI-HSS). Three subscales, Emotional Exhaustion (EE), Depersonalization (DP), and Personal Accomplishment (PA), were calculated by adding up the item scores. Higher scores in EE ($\geq$17) and DP ($\geq$7), and lower scores in PA ($\leq$38), were indicative of a higher level of burnout. The calculated internal reliability of the MBI-HSS, using Cronbach's coefficient alpha, is estimated at 0.89 for EE, 0.79 for DP, and 0.82 for PA [29].

Vaccination readiness was measured using the validated German version of the 5C model [16]. Study participants might answer with a Likert scale from 1 ("I strongly disagree") to 7 ("I strongly agree"). The questionnaire wa evaluated at the item level. High levels of agreement with the item "confidence" and low levels of agreement with the items "complacency", "constraints", "calculation" and "collective responsibility" were associated with an increased vaccination readiness.

### 2.4. Data Sources/Measurement

Each study participant answered pseudonymized paper-based questionnaires individually. As impairments of cognitive and communicative ability (measured by the Six-Item-Screener and Montreal Cognitive Assessment [30,31]) had to be expected among people in need of care, the data collection of self-reported outcomes was ensured according to a pre-specified substitution principle (caregiver/legal representative/HCW). In cases of physical impairment, questionnaires were answered with the support of a trained and qualified study assistant.

### 2.5. Bias

As in most research in outpatient care, the external validity of our findings is vulnerable to participation bias. For example, it is conceivable that non-responding institutions were particularly burdened by the pandemic. Therefore, we provided a mobile study team (including study nurse and study physician), that no additional resources were required to conduct the study. Furthermore, our vaccine-specific interim evaluation was only a small proportion of the topics surveyed. Therefore, it was not apparent to potential study participants that the survey would ask for health literacy or vaccination readiness. Consequently, it has not to be assumed that only study participants in favor of vaccinations were represented in the survey.

### 2.6. Study Size

Sample size calculation was carried out for the prospective register study (BaCoM). Based on 1000 people in need of care, the minimal statistical difference for major outcomes (age, comorbidities, and mortality) between the people in need of care with evidence of a previous SARS-CoV-2 infection, people in need of care without evidence of a previous SARS-CoV-2 infection, and people without need of care but with evidence of a previous SARS-CoV-2 infection was simulated. A two-tailed *t*-test or log-rank test, with the assumptions for the significance level $\alpha$ = 0.05 and the power $\beta$ = 0.8 and the given standard deviation,

was used. Detectable differences for the following variables were obtained: age: SD = 10.0, detectable difference of −2.29 or 2.29; comorbidities: SD = 3.1, detectable difference of −7.10 or 7.10; mortality: median survival time = 4.0, detectable difference of 2.66 or 6.56; EQ-5D-5L: SD = 0.29, detectable difference of −0.07 or 0.07. With respect to the limited life expectancy of care recipients, it was estimated that after four years, about one-third of the study participants would still be alive across all levels of care [22].

### 2.7. Quantitative Variables

Initially, all primary data were recorded in paper-based case report forms and transferred to electronic case report forms as part of a double data entry process using the LibreClinica scientific data management system. The data were analyzed using SPSS statistical software version 28 (IBM Corp., Armonk, New York 10504-1722 United States).

### 2.8. Statistical Methods

In this analysis, metric and normally distributed data were presented with mean and standard deviation, while metric and non-normally distributed data were presented with median and Q1–Q3. For categorical data, frequency and percentage were presented.

Univariate and multivariate ordinal logistic regression models were calculated. The assumptions of ordinal regression with regard to multicollinearity and proportional odds were examined using suitable statistical methods (correlation coefficient, full likelihood ratio test).

To evaluate people in need of care, demographic and psychosocial characteristics were considered as independent variables in the ordinal regression models (outcome health literacy (HLS-EU-Q16): age, sex, marital status (not married/widowed/married), ethnic origin (Caucasian/other), education (non-academic degree/academic degree), type of care (inpatient/outpatient), Barthel Index (Score 0–30/35–80/85–95/100), legal representative (yes/no), symptoms of depression (PHQ9 score metric or ≥10), and symptoms of anxiety disorder (GAD7 score metric or ≥10)).

Demographic and psychosocial characteristics were considered as independent variables in the ordinal regression models to evaluate HCWs as well (outcome vaccination readiness (5C): age, sex, marital status (not married/widowed/married), ethnic origin (Caucasian/other), education (non-academic degree/academic degree), type of care (inpatient/outpatient), symptoms of burnout (MBI emotional exhaustion, MBI depersonalization, MBI personal accomplishment), function in the facility (nursing staff/elderly care staff), employment relationship (full-time employment/part-time employment), and care for COVID-19-infected patients (yes/no)).

All independent variables that were significant in the univariate models (*p*-value < 0.05) were included in the multivariate model. Dependent variables were health literacy (HLS-EU-Q16) for people in need of care and items indicating vaccination readiness (Confidence, Complacency, Constraints, Calculation, and Collective Responsibility (5C)) among HCWs.

The effect of independent variables on the dependent outcome variables (health literacy (HLS-EU-Q16) or vaccination readiness (5C)) was expressed as odds ratios (ORs) with 95% confidence intervals (CIs). The significance level was set at α = 0.05. Missing data are indicated on the item level.

## 3. Results

### 3.1. Participants

A total of *n* = 505 study participants could be included in the cross-sectional analysis, including *n* = 285 people in need of care (56.4%) and *n* = 220 HCWs (43.6%) of *n* = 66 inpatient and outpatient care facilities (see Figure 1). Since potential study participants were recruited from a variety of settings and trajectories (see Section 2.2), it is not possible to report numbers of individuals at each stage of study (e.g., numbers potentially eligible, examined for eligibility, confirmed eligible) or reasons for non-participation. Follow-ups were not part of this cross-sectional analysis.

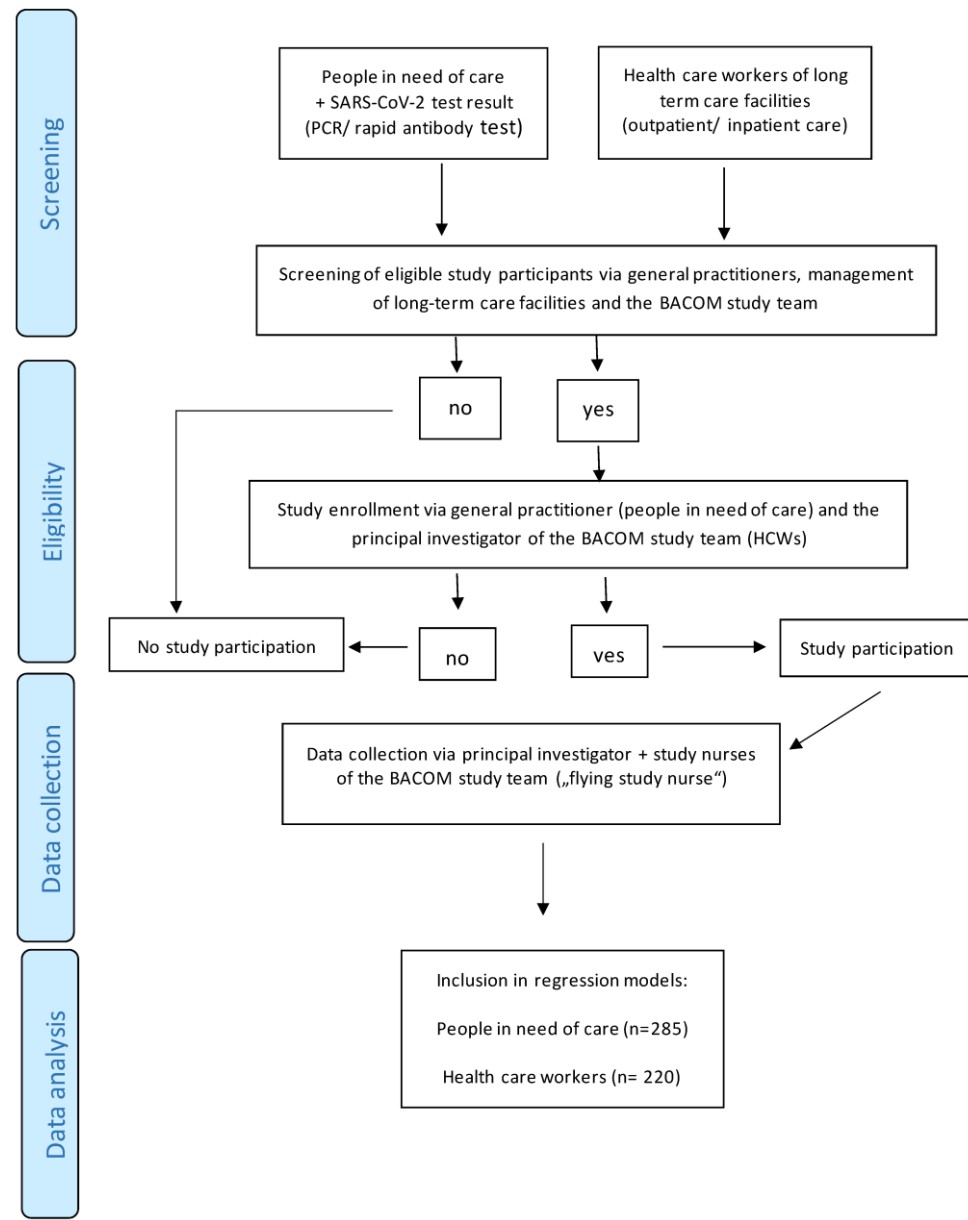

**Figure 1.** Flowchart of data.

*3.2. Descriptive Data, Outcome Data, and Main Results*

3.2.1. People in Need of Care

The median (Q1–Q3) age of the people in need of care was 84.0 (75.0–89.0) years. Although all respondents were in need of care, the majority were still able to carry out many activities of everyday life independently (Barthel Index score median 75.0 (50.0–90.0). In addition, 44.9% (*n* = 128) reported depressive symptoms, of which 14.7% (*n* = 42) were clinically relevant (PHQ-9 score of ≥10). Clinically relevant signs of generalized anxiety disorders (GAD-7 score ≥ 10) were reported by 8.4% (*n* = 24). A total of 37.9% (*n* = 108) showed an insufficient level of health literacy (see Table 1).

A multivariate ordinal regression model indicated that symptoms of major depression (PHQ-9 score ≥10) were associated with lower levels of health literacy (*p* ≤ 0.001 ***). Symptoms of a general anxiety disorder did not show any association with health literacy. The need for a legal representative was also significantly associated with low levels of health literacy (*p* ≤ 0.001 ***), as well as an intermediate Barthel Index Score between 35 and 80 (*p* = 0.042 *). However, having an academic degree had a significantly positive effect on health literacy (*p* = 0.003 **) (see Table 2).

**Table 1.** Characteristics of the people in need of care (*n* = 285).

| Sociodemographic Data | People in Need of Care (*n* = 285) | |
|---|---|---|
| **Age (year), Median (Q1–Q3)** | 84.0 (75.0–89.0) | |
| Missing Data | 2 (0.7%) | |
| Female sex, *n* (%) | 195 (68.4%) | |
| Missing Data | 0 (0.0%) | |
| Marital Status, *n* (%) | | |
| Not Married/Widowed | 222 (77.9%) | |
| Married | 63 (22.1%) | |
| Missing Data | 0 (0.0%) | |
| Ethnic Origin, *n* (%) | | |
| Caucasian | 283 (99.3%) | |
| Others | 1 (0.4%) | |
| Missing Data | 1 (0.4%) | |
| Education, *n* (%) | | |
| Non-academic degree | 225 (78.9%) | |
| Academic degree | 53 (18.6%) | |
| Missing Data | 7 (2.5%) | |
| Type of Care, *n* (%) | | |
| Inpatient | 198 (69.5%) | |
| Outpatient | 66 (23.2%) | |
| Missing Data | 21 (7.4%) | |
| Barthel Index Score, Median (Q1–Q3) | 75.0 (50.0–90.0) | |
| Missing Data | 2 (0.7%) | |
| Legal Representative, *n* (%) | | |
| Yes | 114 (40.0%) | |
| No | 161 (56.5%) | |
| Missing Data | 10 (3.5%) | |
| **Scale** | **n (%)** | **Median (Q1–Q3)** |
| Score PHQ-9 (4-Scale) | 282 (98.9%) | 4.0 (2.0–7.0) |
| PHQ-9, depression symptoms | | |
| No depression symptoms | 154 (54.0%) | |
| Mild depression symptoms | 86 (30.2%) | |
| Moderate depression symptoms | 30 (10.5%) | |
| Moderate to severe depression symptoms | 9 (3.2%) | |
| Severe depression symptoms | 3 (1.1%) | |
| Missing | 3 (1.1%) | |
| Clinical depression—PHQ-9 score $\geq$ 10 | 42 (14.7%) | |
| Score GAD-7 (4-Scale) | 277 (97.2%) | 1.0 (0.0–4.0) |
| GAD-7, general anxiety disorder | | |
| No anxiety disorder | 216 (75.8%) | |
| Mild anxiety disorder | 37 (13.0%) | |
| Moderate anxiety disorder | 18 (6.3%) | |
| Severe anxiety disorder | 6 (2.1%) | |
| Missing | 8 (2.8%) | |
| Clinical anxiety—GAD-7 score $\geq$ 10 | 24 (8.4%) | |
| Score HLS-EU-Q16 (2-Scale) | 285 (100.0%) | 14.0 (10.0–16.0) |
| HLS-EU-Q16, health literacy | | |
| Inadequate | 55 (19.3%) | |
| Problematic | 53 (18.6%) | |
| Sufficient | 177 (62.1%) | |

**Table 2.** Univariate and multivariate ordinal logistic regression analysis of people in need of care.

| People in Need of Care | | Health Literacy HLS-EU_Q16 | |
|---|---|---|---|
| | | **Univariate (Odds Ratio (95%CI); *p*-Value)** | **Multivariate (Odds Ratio (95%CI); *p*-Value)** |
| Martial Status | | 2.0 (1.1–3.8); 0.023 * | 1.8 (0.9–3.5); 0.101 |
| Academic Degree | | 2.2 (1.1–4.3); 0.024 * | 3.1 (1.5–6.6); 0.003 ** |
| Barthel Index Score | 0–30 | 0.4 (0.2–1.0); 0.051 | 0.5 (0.2–1.3); 0.151 |
| | 35–80 | 0.4 (0.2–0.9); 0.020 * | 0.4 (0.2–1.0); 0.042 * |
| | 85–95 | 0.6 (0.3–1.4); 0.233 | 0.9 (0.4–2.1); 0.736 |
| Legal Representative | | 0.4 (0.2–0.9); <0.001 *** | 0.4 (0.2–0.6); <0.001 *** |
| PHQ9 Clinical Cut-off $\geq$ 10 | | 0.4 (0.2–0.7); <0.001 *** | 0.3 (0.2–0.6); 0.001 *** |

Legend *: $p \leq 0.05$, **: $p \leq 0.01$, ***: $p \leq 0.001$.

### 3.2.2. Healthcare Workers (HCWs)

The median (Q1–Q3) age of the HCWs was 47.0 (34.0–56.0) years, and 76.8% (*n* = 169) were employed full-time. More than half (60.5%, *n* = 133) of the HCWs reported signs of depressive syndromes, of which 28.2% (*n* = 62) were clinically relevant (PHQ-9 score of $\geq$10). Many HCWs had symptoms of burnout; 21.4% (*n* = 47) showed a high level of MBI-EE, 17.3% (*n* = 38) showed high levels of DP, and 55.0% (*n* = 121) showed high levels of PA. Concerning vaccination readiness, the HCWs showed high levels of Confidence with a median of (Q1–Q3) 4.0 (2.0–5.0). They showed low levels of Complacency 1.0 (0.0–3.0) and indicated Constraints in some cases 0.0 (0.0–2.0). The Calculation score was high 4.0 (2.0–5.0). Collective Responsibility was very low 0.0 (0.0–2.0), which indicates a high level of vaccination readiness, due to inverse phrasing of this item. A total of 73.6% (*n* = 162) of the HCWs reported caring for COVID-19-infected people in need of care since the beginning of the pandemic (see Table 3).

**Table 3.** Characteristics of the healthcare workers (HCWs) (*n* = 220).

| Sociodemographic Data | Healthcare Workers (HCWs) (*n* = 220) |
|---|---|
| **Age (year), Median (Q1–Q3)** | 47.0 (34.0–56.0) |
| Missing Data | 5 (2.3%) |
| Female sex, *n* (%) | 174 (79.1%) |
| Missing Data | 0 (0.0%) |
| Marital Status, *n* (%) | |
| Not Married/Widowed | 130 (59.1%) |
| Married | 90 (40.9%) |
| Missing Data | 0 (0.0%) |
| Ethnic Origin, *n* (%) | |
| Caucasian | 191 (86.8%) |
| Others | 19 (8.6%) |
| Missing Data | 10 (4.5%) |
| Education, *n* (%) | |
| Non-academic degree | 139 (63.2%) |
| Academic degree | 79 (35.9%) |
| Missing Data | 2 (0.9%) |
| Type of Care, *n* (%) | |
| Inpatient | 215 (97.7%) |
| Outpatient | 5 (2.3%) |
| Missing Data | 0 (0.0%) |

**Table 3.** *Cont.*

| Sociodemographic Data | Healthcare Workers (HCWs) (*n* = 220) | |
|---|---|---|
| Function in the Facility, *n* (%) | | |
| Nursing Staff | 31 (14.1%) | |
| Elderly Care Staff | 143 (65.0%) | |
| Missing Data | 46 (20.9%) | |
| Employment Relationship, *n* (%) | | |
| Full-time employed | 169 (76.8%) | |
| Part-time employed | 49 (22.3%) | |
| Missing Data | 2 (0.9%) | |
| Care for COVID-19-infected patients, *n* (%) | 162 (73.6%) | |
| Missing Data | 12 (5.5%) | |
| **Scale** | ***n* (%)** | **Median (Q1–Q3)** |
| Score PHQ-9 (4-Scale) | 217 (98.6%) | 6.0 (3.0–10.0) |
| PHQ-9, depression Syndromes, No. (%) | | |
| No depression Syndromes | 84 (38.2%) | |
| Mild depression Syndromes | 71 (32.3%) | |
| Moderate depression Syndromes | 44 (20.0%) | |
| Moderate to severe depression Syndromes | 13 (5.9%) | |
| Severe depression Syndromes | 5 (2.3%) | |
| Missing | 3 (1.4%) | |
| Clinical depression—score $\geq$ 10 | 62 (28.2%) | |
| MBI total score (7-Scale) | | |
| Emotional Exhaustion (EE) | 204 (92.7%) | 15.5 (9.0–26.0) |
| Low | 106 (48.2%) | |
| Average | 51 (23.2%) | |
| High | 47 (21.4%) | |
| Missing | 16 (7.3%) | |
| Depersonalization (DP) | 217 (98.6%) | 5.0 (1.5–9.0) |
| Low | 131 (59.5%) | |
| Average | 48 (21.8%) | |
| High | 38 (17.3%) | |
| Missing | 3 (1.4%) | |
| Personal Accomplishment (PA) | 214 (97.3%) | 30.0 (23.0–36.0) |
| Low | 34 (15.5%) | |
| Average | 59 (26.8%) | |
| High | 121 (55.0%) | |
| Missing | 6 (2.7%) | |
| Vaccination readiness 5C (7-Scale) | | |
| Confidence | 219 (99.5%) | 4.0 (2.0–5.0) |
| Complacency | 219 (99.5%) | 1.0 (0.0–3.0) |
| Constraints | 218 (99.1%) | 0.0 (0.0–2.0) |
| Calculation | 219 (99.5%) | 4.0 (2.0–5.0) |
| Collective Responsibility | 197 (89.5%) | 0.0 (0.0–2.0) |

A multivariate ordinal regression model indicated that caring for COVID-19-infected patients was negatively associated with Confidence in vaccinations (*p* = 0.016). Symptoms of depression did affect vaccination readiness in HCWs, as the perception of Constraints was significantly associated with the PHQ-9 Score (*p* = 0.023 *). A feeling of DP did increase Complacency levels (*p* = 0.005) as well as a perception of Constraints towards vaccinations (*p* = 0.010). Collective responsibility to protect others from vaccine-preventable diseases was reduced significantly in HCWs suffering from burnout symptoms (*p* = 0.001). A higher educational level was associated with an increased level of Calculation (*p* = 0.034; see Table 4).

**Table 4.** Univariate and multivariate ordinal logistic regression analysis of healthcare workers (HCWs).

| Healthcare Workers (HCWs) | Univariate (Odds Ratio (95%CI); *p*-Value) | Multivariate (Odds Ratio (95%CI); *p*-Value) |
|---|---|---|
| **Confidence** | | |
| Current care for COVID-19-infected patients | 0.4 (0.2–0.9); 0.023 * | 0.3 (0.1–0.8); 0.016 * |
| PHQ9 Score | 0.9 (0.9–1.0); 0.006 ** | 0.9 (0.9–1.0); 0.066 |
| MBI—Depersonalization | 0.4 (0.2–0.8); 0.009 ** | 0.6 (0.3–1.2); 0.168 |
| MBI—Personal Accomplishment | 0.4 (0.2–1.0); 0.043 * | 0.6 (0.2–1.4); 0.203 |
| **Complacency** | | |
| MBI—Depersonalization | 2.9 (1.4–6.0); 0.005 ** | 2.9 (1.4–6.0); 0.005 ** |
| **Constraints** | | |
| Current care for COVID-19-infected patients | 2.6 (1.0–6.6); 0.046 * | 2.5 (0.9–7.4); 0.092 |
| PHQ9 Score | 1.1 (1.0–1.2); 0.008 ** | 1.1 (1.0–1.2); 0.023 * |
| MBI—Emotional Exhaustion | 2.4 (1.1–5.4); 0.028 * | 0.7 (0.2–2.0); 0.526 |
| MBI—Depersonalization | 3.9 (1.8–8.6); 0.001 *** | 3.7 (1.4–10.0); 0.010 ** |
| **Calculation** | | |
| Academic | 2.0 (1.1–3.4); 0.016 * | 1.9 (1.0–3.3); 0.034 * |
| MBI—Depersonalization | 0.5 (0.9–0.9); 0.033 * | 0.5 (0.3–1.1); 0.071 |
| MBI—Personal Accomplishment | 0.4 (0.2–1.0); 0.046 * | 0.5 (0.2–1.2); 0.114 |
| **Collective Responsibility** | | |
| MBI—Depersonalization | 4.5 (1.9–10.8); 0.001 *** | 4.5 (1.9–10.8); 0.001 *** |

Legend: *: $p \leq 0.05$, **: $p \leq 0.01$, ***: $p \leq 0.001$.

## 4. Discussion

### 4.1. Key Results

Within this analysis, some psychosocial factors could be revealed that might be associated with actual vaccination behavior in the setting of long-term care. People in need of care with clinically relevant depressive symptoms showed signs of a significantly reduced health literacy. Among HCWs, symptoms of depression and burnout were negatively associated with vaccination readiness. Confidence in vaccinations was significantly reduced among HCWs caring for COVID-19-infected patients.

### 4.2. Strength and Limitations

Under very dynamic and often restrictive conditions, this study captured the psychosocial situation of people in need of care and HCWs on the frontline of an ongoing pandemic. Despite cognitive and physical impairments, it was possible to capture the emotional world and socio-demographic data of particularly vulnerable groups. However, cross-sectional data from a registry study are of limited use in the context of a pandemic, as it could not map the course of a population's psychosocial antecedents during the pandemic and only provided a snapshot. As people in need of care suffer from a broad range of physical diseases and impairments, these might have affected mental health issues of this study population as well as the COVID-19 pandemic did. In addition, the vaccination rates for both target groups have not yet been fully recorded. Therefore, it is not possible to draw any conclusions about vaccination behavior.

### 4.3. Interpretation

Concerning people in need of care, a strong association between signs of clinically relevant depression and their health literacy could be examined. A high level of education was identified as a promoting factor for sufficient health literacy in the elderly, which is consistent with existing literature [7]. These associations were of particular significance in the present analysis, as these data were collected during the ongoing COVID-19 pandemic with actively implemented hygiene measures. Community-dwelling elderly before 2019 did show less significant associations [7]. A possible explanation for this relation might be the mediating roles of social support and depression between health literacy and frailty in the elderly. People in need of care who have stronger social support seem to be less frail [32]. In addition, physical and psychological support can help to improve the quality of life of older people and help to alleviate frailty and depression [33]. Better medical conditions can be provided and more social interactions can be offered by effective social support without pandemic restrictions. Moreover, the elderly can develop emotional comfort from social networks to alleviate their negative feelings and nurture their ability more effectively to deal with their own frailty [34]. Therefore, providing sufficient social support represents an effective strategy to mitigate frailty as well as depressive symptoms in the elderly.

Among HCWs, high levels of depressive symptoms and burnout were detected.

Stress and burnout are associated with substance abuse, chronical illness, anxiety and depression, and they lead in many cases to complacency as a protection factor [35]. Automated task completion and a basic level of complacency may also be important to avoid further deterioration in mental health. In the context of vaccination readiness, complacency might be dangerous, since it is associated with reduced vaccination rates [16,36]. However, it seems consistent that people with high levels of stress and burnout show reduced levels of risk perception in terms of vaccine-preventable diseases. Furthermore, symptoms of depression, burnout, and especially depersonalization are associated with subjectively perceived organizational constraints at work [37]. Constraints to get vaccinated might also be affected.

Depersonalization is a symptom of burnout characterized by a sense of alienation from oneself, others and the environment. It can cause sufferers to feel emotionally distant and disconnected from those around them. Indeed, studies show that depersonalization is an important factor affecting people's empathy levels [29,35]. The results of this analysis showed that the dimension of depersonalization has a significant negative association with Collective Responsibility.

Although COVID-19 vaccines have shown excellent clinical efficacy and effectiveness in real-world data, vaccine breakthroughs have led to vaccination hesitancy in some people [38]. This observation can be supported by the findings of this study, as Confidence in the safety and effectiveness of available COVID-19 vaccines was lower in HCWs who have cared for COVID-19-infected patients.

A higher level of education seems to have a positive correlation with a conscious evaluation of vaccination information in the studied population of HCWs (Calculation). This finding is supported by previous studies that have shown that the educational level might influence vaccination readiness and emphasizes the importance of health facilities providing accurate and easily accessible information about COVID-19 vaccination [39].

### 4.4. Implications for Research and Practice

Group activities and good social connection can help to alleviate depressive symptoms and thus improve mental health outcomes as well as health literacy in people in need of care according to individual possibilities in the setting of long-term care [7]. The involvement of family members and informal caregivers could also be a promising tool to reduce the severity of depressive episodes [40]. Family engagement as part of managing patients with mental illness seems to be useful in terms of improved compliance with medication and treatment plans as well [41]. Educational interventions might be helpful to improve preventive behavior in those people. Systemic assistance and interventions specialized

for frail elderly in long-term care facilities and their caregivers need to be developed and tested to improve clinical practice and patient health literacy [42]. Above all, greater involvement of relatives and informal caregivers of mentally ill people should be intensified and systematically evaluated.

To improve vaccination readiness among HCWs, targeted educational work could reduce misinformation as well as educate about disease consequences, vaccination risks, and vaccination as a collective decision [43]. Vaccine-centered training should be integrated in the curriculum and in occupational training of HCWs. Incentivization with training points or certificates could further strengthen the acceptance of this measure. To further address HCWs' concerns about vaccine safety, it is suggested to improve public health message support and promote vaccination [44]. It should also be emphasized that vaccination is for the protection of the HCWs themselves, their patients, as well as their families and friends [43]. Above all, to mitigate the problem of burnout among HCWs, it is essential to eliminate the personnel bottlenecks in care and to find ways to strengthen resilience in healthcare teams to provide and maintain safe patient care [45].

### 4.5. Generalizability

Within this cross-sectional analysis, we did not aim to measure incidences of mental health issues or to make any causal inference. Furthermore, it has to be considered that there might be strong temporal and regional influence on vaccination readiness due to the dynamics of an ongoing pandemic. It is also unclear how non-responders might have answered this survey. To understand better who does and does not participate, we will conduct an analysis of a subsample of non-responding care facilities scheduled 6–12 months after the first contact, in order to elicit structural and contextual information about the facilities [22].

However, lack of resources in long-term care affects almost all healthcare systems in Organization for Economic Cooperation and Development (OECD) countries [46]. As the global event of the COVID-19 pandemic had comparable effects on the mental health of people in need of care and HCWs worldwide [12,13,36], it can be assumed that the practical implications of this study can be generalized.

### 5. Conclusions

The results of the present study suggest that people in need of care with symptoms of depression and HCWs with symptoms of burnout could benefit from intensified target group-specific vaccination counseling. Among people in need of care, their individual needs, restrictions as well as the importance of their social environment, have to be considered.

In the case of HCWs, symptoms of burnout showed a complex relationship with psychological antecedents of vaccination. This phenomenon should be further investigated and appropriate measures for burnout prevention at a political and structural level are needed.

**Supplementary Materials:** The following supporting information can be downloaded at: https://www.mdpi.com/article/10.3390/ejihpe14030029/s1. STROBE Statement—checklist.

**Author Contributions:** Conceptualization, L.S., T.D. and J.G.; Methodology, L.S.; interpretation of results, L.S.; Validation, M.G.; Investigation, M.G., A.G., I.Z., and C.J.; Resources, T.K., I.G., A.H., M.H., D.T., T.D. and J.G.; Data acquisition, M.G., A.G., I.Z., M.S., D.W., C.J. and M.R.; Data analysis: M.S., A.G. and M.R.; Supervision, L.S., T.K., I.G., A.H., M.H., D.T., T.D. and J.G.; Project Administration, M.G., A.G., I.Z. and C.J.; Funding Acquisition, T.D. and J.G. All authors have read and agreed to the published version of the manuscript.

**Funding:** This work is part of the "Bayerischer ambulanter COVID-19 Monitor-BaCoM" and was supported by the Bavarian State Ministry of Health and Care (grant number: G45a-G8300-2021/257-2).

**Institutional Review Board Statement:** Ethics approval and consent to participate: All procedures contributing to this work comply with the ethical standards of the relevant national and institutional committees on human experimentation and with the Helsinki Declaration of 1975, as revised in

2008. All procedures involving human patients were approved by the ethics committee of the Ludwigs-Maximilians-University in Munich under the case number 20-0860.

**Informed Consent Statement:** All participants provided informed written consent for their participation and the usage of their pseudonymized data.

**Data Availability Statement:** The datasets used and/or analyzed during the current study are available from the corresponding author on reasonable request.

**Acknowledgments:** We would like to thank all participants of the BaCoM study, especially the people in need of care and healthcare workers who agreed to share with us their personal experiences during an ongoing pandemic. These very personal insights might help to develop and improve long-term care, especially in terms of mental health issues. We also would like to thank the Bavarian State Ministry of Health and Care for support as well as the BaCoM study group. Members of the BaCoM study group are Jochen Gensichen (PI), Tobias Dreischulte, Ildikó Gágyor, Anita Hausen, Michael Hölscher, Christian Janke, Thomas Kühlein, Armin Nassehi, Daniel Teupser, Felix Bader, Barbara Daubner, Christine Eidenschink, Caroline Floto, Dagmar Hindenburg, Sylvi Hoffmann, Peter Kurotschka, Daniela Lindemann, Karoline Lukaschek, Katharina Mayr, Irina Michel, Susan Müller, Marietta Rottenkolber, Linda Sanftenberg, Florian Arend, Jessica Scheel-Bartelt, Rita Schwaiger, Maria Sebastiao, Sabrina Weigand, Domenika Wildgruber, Susanne Winter, Isabel Zöllinger, Heidi Hentschel, Christina Huber, Julian Mayrhuber, Mara Pettke, Sophia Straub, Alexander Theiss. Thanks to the German Center for Mental Health (DZPG) and Bavarian Practice Research network (BayFoNet).

**Conflicts of Interest:** The authors declare that they have no competing interests.

## List of Abbreviations (in Alphabetical Order)

| | |
|---|---|
| BaCoM | Bavarian ambulatory COVID-19 monitor |
| 5C | psychological antecedents of vaccination readiness |
| CFS | clinical frailty scale |
| CI | confidence interval |
| OR | odds ratio |
| DP | depersonalization |
| DRKS | German Register of Clinical Studies |
| DSM-5 | Diagnostic and Statistical Manual of Mental Disorders 5th edition |
| EE | emotional exhaustion |
| GAD-7 | Generalized Anxiety Disorder Scale-7 |
| GP | general practitioner |
| HCWs | healthcare workers |
| HLS-EU-Q16 | Health Literacy Questionnaire |
| IQR | interquartile range |
| MBI-HSS | Maslach Burnout Inventory—Human Services Survey |
| OECD | Organization for Economic Cooperation and Development |
| PA | personal accomplishment |
| PCR | polymerase-chain reaction |
| PHQ-9 | Patient Health Questionnaire-9 |
| Q | quartile |

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
