# Peer review of "Associations of Mental Health Issues with Health Literacy and Vaccination Readiness against COVID-19 in Long-Term Care Facilities—A Cross-Sectional Analysis"

_ejihpe, doi:10.3390/ejihpe14030029_

Round 1

Reviewer 1 Report

Comments and Suggestions for Authors

This is a well performed and presented study of the influence of mental health issues on the performance of Covid-19 vaccination in patients and health care workers in long-term facilities, and points to issues to adress to promote the vaccination program. 

I have no suggestions for major changes to be made, only a few minor things:

line 141: remove the hyphen in questionnaire

Insert an empty line over each of the Table headings, lines165,172, 186, 197

In Table 4, the cell for MBI-Depersonalization vs Constraints has a p-value higher than the set significance level and should not have the green background and the two starlets indicating p<0.01

line207: people

Reviewer 2 Report

Comments and Suggestions for Authors

Vaccinations against Covid-19 are of utmost importance in long-term care facilities.

 During the pandemic, mental health issues increased and might influence health literacy as well as vaccination readiness.

AUTHORS aimed to assess the influence of depression and anxiety on health literacy in people in need of care and the influence of depression and burnout on vaccination readiness against Covid-19 in health care workers (HCWs).

Within THEIR cross-sectional study, people in need of care were asked for symptoms of depression (PHQ-9), anxiety (GAD-7) and health literacy (HLS-EU-Q16).

Among HCWs, THEY assessed symptoms of depression (PHQ-9) and burnout (MBI-HSS) as well as psychological antecedents of vaccination (5C) to measure vaccination readiness against Covid-19. Multivariate regression analysis with stepwise selection of variables was performed.

Symptoms of a major depression were significantly associated with reduced health literacy (p=0.010) in people in need of care. Among HCWs, symptoms of burnout reduced vaccination readiness against Covid-19 significantly. In particular, collective responsibility was reduced in HCWs suffering from burnout symptoms (p<0.001).

People in need of care and their HCWs could benefit from intensified target group specific vaccination counseling. Nonetheless, more attention should be paid to the protection of mental health in long-term care facilities.

This is an interesting study.

I have some minor suggestions for the authors:

1.       There are a lot of acronyms (also in the abstract) that must be defined before. Please add a lsit

2.       Use [] when you cite references

3.       Better define the purpose, using bullet points to better specify the type of study.

4.       Introduce  the themes of the results by means of a few sentences.

5.       If possible avoid using “we” and “our “

Reviewer 3 Report

Comments and Suggestions for Authors

Thank you for the opportunity to review an interesting topic, namely, Impact of mental health issues on health literacy and vaccination readiness against Covid-19 in long-term care facilities – a cross sectional analysis.

I have major concerns, as follows:

1. Lines 80-82: “Therefore, the aim of this study was to assess the influence of depression and anxiety on health literacy in people in need of care and the influence of depression and burnout on vaccination readiness against Covid-19 in health care workers”.

I suggest that authors write only about symptoms of mental health disorders. This study was a single cross-sectional in design, hence the terms such as influence, impact, determinants, etc. cannot be written. I suggest using terms, as follows: association, correlation, relationship, etc.

2. As the Authors carried out a single cross-sectional study, the sample size of 285 people was very small as well as non-representative.

3. The sampling procedure remains unclear. Therefore, the Authors should add the recruitment process of the study participants; the authors should add the inclusion and exclusion criteria of this study; did all the study participants have no diseases? The health status of the study participants can affect the mental health; the authors should add the methods for calculating the sample size; according to the recruitment process of this study, the authors should add a flow diagram of this study.

4. Statistical analysis section could provide more information.

5. Table 1:  The Authors only write about depressive symptoms. It is not correct to use the term “syndromes”.

6. Some sentences are incomprehensible and confusing: “As expected, the need for a legal representative was also significantly associated with low levels of health literacy (lines 168-169)”.

7. Did the Authors apply the logistical regression analysis method for data analysis? If yes, it is necessary to specify the odds ratios. What fitting criteria were used by the Authors for regression models?

8. The generalizability (and implications for the potential readers in foreign countries) of this study should be added.

9. References must be adjusted to citation style.

Comments on the Quality of English Language

English very difficult to understand/incomprehensible.

Round 2

Reviewer 3 Report

Comments and Suggestions for Authors

The Authors changed the manuscript in part. Unfortunately, this is not enough. I am updating further additional comments and concerns.

Considering that the Authors carried out a single cross-sectional study, I found the manuscript interesting. However, I  have a revision request. I suggest the Authors use STROBE checklist in reporting their cross-sectional study. Please, add the STROBE checklist as a supplementary material and cite it through the main text. You can download the checklist from this link https://www.strobe-statement.org/.

Also, I have concerns concerning the design of the study, the organisation of the study, and the description of the Methodology as well as Results. These concerns in the manuscript can be answered in accordance with the STROBE checklist.

Round 3

Reviewer 3 Report

Comments and Suggestions for Authors

The authors partially corrected the manuscript.

Nevertheless, I do not recommend the acceptation of the paper.

The main drawbacks of the manuscript relate to the design of the study and statistical data analysis. The authors carried out a mass of statistical tests to verify the null hypothesis.

The authors carried out a multivariate logistic regression analysis and concluded a multivariate ordinal regression model, either.

However, according to the design of the single cross-sectional study, it remains unclear which variables were independent (or risk factors) and which variable was dependent (an outcome). It is also unclear which independent variables were attributed to confounders.

Which statistical indicators have shown that the logistic regression model was appropriate for prognosis?

I suggest that authors repeat the organization of the study as well as repeat the statistical data analysis. As a consequence, the results and conclusions of the study will change.
